# Skin Barrier Function in Psoriasis and Atopic Dermatitis: Transepidermal Water Loss and Temperature as Useful Tools to Assess Disease Severity

**DOI:** 10.3390/jcm10020359

**Published:** 2021-01-19

**Authors:** Trinidad Montero-Vilchez, María-Victoria Segura-Fernández-Nogueras, Isabel Pérez-Rodríguez, Miguel Soler-Gongora, Antonio Martinez-Lopez, Ana Fernández-González, Alejandro Molina-Leyva, Salvador Arias-Santiago

**Affiliations:** 1Dermatology Department, Hospital Universitario Virgen de las Nieves, Avenida de Madrid, 15, 18012 Granada, Spain; tmonterov@correo.ugr.es (T.M.-V.); antoniomartinezlopez@aol.com (A.M.-L.); salvadorarias@ugr.es (S.A.-S.); 2Instituto de Investigación Biosanitaria GRANADA, 18012 Granada, Spain; ana.fernandez.gonzalez@juntadeandalucia.es; 3Dermatology Department, Faculty of Medicine, University of Granada,18001 Granada, Spain; victoriasfn@correo.ugr.es (M.-V.S.-F.-N.); isabelpr@correo.ugr.es (I.P.-R.); miguelsg@correo.ugr.es (M.S.-G.)

**Keywords:** atopic dermatitis, homeostasis, psoriasis, skin barrier, transepidermal water loss

## Abstract

Multiple diagnostic tools are used to evaluate psoriasis and atopic dermatitis (AD) severity, but most of them are based on subjective components. Transepidermal water loss (TEWL) and temperature are skin barrier function parameters that can be objectively measured and could help clinicians to evaluate disease severity accurately. Thus, the aims of this study are: (1) to compare skin barrier function between healthy skin, psoriatic skin and AD skin; and (2) to assess if skin barrier function parameters could predict disease severity. A cross-sectional study was designed, and epidermal barrier function parameters were measured. The study included 314 participants: 157 healthy individuals, 92 psoriatic patients, and 65 atopic dermatitis patients. TEWL was significantly higher, while stratum corneum hydration (SCH) (8.71 vs. 38.43 vs. 44.39 Arbitrary Units (AU)) was lower at psoriatic plaques than at uninvolved psoriatic skin and healthy controls. Patients with both TEWL > 13.85 g·m^−2^h^−1^ and temperature > 30.85 °C presented a moderate/severe psoriasis (psoriasis area severity index (PASI) ≥ 7), with a specificity of 76.3%. TEWL (28.68 vs. 13.15 vs. 11.60 g·m^−2^ h^−1^) and temperature were significantly higher, while SCH (25.20 vs. 40.95 vs. 50.73 AU) was lower at AD eczematous lesions than uninvolved AD skin and healthy controls. Patients with a temperature > 31.75 °C presented a moderate/severe AD (SCORing Atopic Dermatitis (SCORAD) ≥ 37) with a sensitivity of 81.8%. In conclusion, temperature and TEWL values may help clinicians to determine disease severity and select patients who need intensive treatment.

## 1. Introduction

The skin is the largest organ of the human body and accomplishes multiple defensive and regulatory functions [1]. The barrier function of skin resides in the epidermis, mainly in the stratum corneum [2]. This epidermal barrier maintains cutaneous homeostasis and protects the body against numerous external stressors [3]. Assessment of epidermal barrier function usually involves measurements of transepidermal water loss (TEWL) [4], stratum corneum hydration (SCH) [5], skin surface pH [6], temperature [7], elasticity [8], melanin [9], and erythema index [10].

Psoriasis and atopic dermatitis (AD) are cutaneous inflammatory diseases resulting from the interaction between environmental and genetic factors that may alter epidermal barrier function [11]. Hyper-proliferation and defective keratinocyte differentiation in psoriasis [12] and decreased filaggrin expression [13] may impair epidermal barrier function. There are scarce reports regarding barrier function characteristics in psoriasis and atopic dermatitis [14,15]. Nevertheless, the assessment of skin homeostasis and epidermal barrier function in these diseases could evaluate qualitative and quantitative skin alterations of lesioned and non-lesioned skin and help to understand the complex and still incomplete etiopathogenesis of these diseases [15].

Moreover, multiple diagnostic tools have been used to evaluate severity in patients with psoriasis and AD [16,17]. The psoriasis area severity index (PASI) is the most widely used scale for assessing psoriasis severity [18]. This score quantifies extent (the percentage of involvement of the four anatomical regions: head, trunk, and upper and lower extremities) and intensity of the psoriatic plaques (evaluating erythema, desquamation, and induration separately for the four anatomical regions) [19]. The SCORing Atopic Dermatitis (SCORAD) is the most common index used to assess AD severity [20]. It consists of the evaluation of the extent of the disorder, the intensity (composed of six items: erythema, oedema/papules, effect of scratching, oozing/crust formation, lichenification, and dryness) and subjective symptoms (itch, sleeplessness) [21]. In therapeutics and outcome research, it is important to measure psoriasis and AD severity, but all of these scales have a subjective component that could lead to a high intra- and inter-observer variability [22,23]. In that way, the measurement of skin homeostasis and epidermal barrier function in psoriatic and AD patients could help clinicians to assess the disease severity objectively [24].

Thus, the objectives of this study are (1) to compare cutaneous homeostasis and skin barrier function between healthy skin, psoriatic skin, and AD skin; and (2) to assess if skin homeostasis and skin barrier function could predict disease severity.

## 2. Materials and Methods

### 2.1. Design

A cross-sectional study was undertaken to assess skin homeostasis differences between healthy skin; involved and uninvolved skin in psoriatic patients; and involved and uninvolved skin in AD patients.

### 2.2. Study Population

Participants were recruited from October 2019 to February 2020 in the Dermatology Service of the Hospital Universitario Virgen de las Nieves in Granada.

Inclusion Criteria:Healthy volunteers were people who attended the Dermatology Service for common conditions, such as melanocytic nevi or seborrheic keratoses, and did not have previous personal or family history of any inflammatory skin disease.Patients with psoriasis were patients with an established clinical diagnosis of mild to severe plaque-type psoriasis [25] and had a psoriasis plaque on their elbows.Patients with AD were patients with established clinical diagnosis of mild to severe AD [26] and had an eczematous lesion on their volar forearms.

Exclusion Criteria:Psoriasis patients currently having non-plaque forms of psoriasis, e.g., erythrodermic, guttate, or pustular psoriasis, or a drug-induced form of psoriasis.Healthy volunteers who had previous personal history of any inflammatory skin disease.Clinical infection on the measured area.History of cancer, including skin cancer.Subjects with intense sun exposure during the study.Not signing the informed consent form.

### 2.3. Study Variables

Main variables of interest

Homeostasis parameters related to epidermal barrier function were measured. SCH (in arbitrary units, using Corneometer^®^ CM 825, Mirocaya, Bilbao, Spain), TEWL (in g·m^−2^ h^−1^, using Tewameter^®^ TM 300, Mirocaya, Bilbao, Spain), pH (using Skin-pH-Meter^®^ PH 905, Mirocaya, Bilbao, Spain), erythema and melanin index (in arbitrary units, using Mexameter^®^ MX 18, Mirocaya, Bilbao, Spain), skin temperature (in °C, using Skin-Thermometer ST 500, Mirocaya, Bilbao, Spain), and elasticity parameters (including R2 value, measured in %, using Cutometer^®^ Dual MPA 580, Mirocaya, Bilbao, Spain) were measured by a Multi Probe Adapter (MPA, Courage + Khazaka electronic GmbH, Mirocaya, Bilbao, Spain). Elasticity parameters were measured four times and the other variables were measured ten times, using their average for analysis. All of these measurements were taken following the same order. All measurements were taken in the same room at a mean room temperature of 23 ± 1 °C and ambient air humidity of 45% (range, 40–50%). All participants underwent an adaptation period of at least 20 min before the measurements were taken. No systemic or topical treatments was allowed three hours before the measurements were taken.

These variables were measured at two body sites in psoriatic patients (on a psoriatic plaque and on an uninvolved skin area at the elbow), at two body sites in AD patients (on an eczematous lesion and on an uninvolved skin area at volar forearm), at one body site in healthy volunteers (on the elbow in controls for psoriasis or on the volar forearm in controls for atopic dermatitis).

Other variables of interest

Data were gathered in a clinical interview on the participants’ sex, age, smoking/alcohol habits, family history of cutaneous disease, personal history of cutaneous disease, skincare habits (moisturizing or suntan lotion use), and hours of sun exposure during the previous week. Psoriasis severity was assessed by the PASI and body surface area (BSA), and AD severity was assessed by SCORAD.

### 2.4. Outcome Measures

Primary outcome measures:To assess differences in TEWL, SCH, and temperature values between healthy skin, psoriatic skin, and AD skin.To evaluate TEWL and temperature values’ ability to discriminate mild psoriasis versus moderate/severe psoriasis.To evaluate TEWL and temperature values’ ability to discriminate mild AD versus moderate/severe AD.

Secondary outcome measures:
To assess differences in other homeostasis parameters between healthy skin, psoriatic skin, and AD skin: erythema, melanin, pH, and elasticity.To assess differences in homeostasis parameters between mild psoriasis and moderate/severe psoriasis: TEWL, SCH, temperature, erythema, melanin, pH, and elasticity.To assess differences in homeostasis parameters between mild AD and moderate/severe AD: TEWL, SCH, temperature, erythema, melanin, pH, and elasticity.

### 2.5. Statistical Analysis

In a descriptive analysis, continuous variables were expressed as means ± standard deviations (SDs) and qualitative variables as absolute and relative frequency distributions. The Student’s *t*-test for independent samples or Student’s *t*-test for paired samples, as appropriate, was used for comparisons of continuous variables. The Pearson correlation coefficient was calculated to test for possible correlations between continuous variables. AD severity was analyzed to establish cut-off points using receiver operating characteristic curves ROC curves for the values of temperature, TEWL, and SCH. The results of ROC curves were used to calculate sensitivity and specificity for various criteria together. Statistical significance was defined by a two-tailed *p* < 0.05. SPSS version 24.0 (SPSS Inc, Chicago, IL, USA) was used for statistical analyzes.

## 3. Results

The study included 314 participants, consisting of 92 patients with psoriasis and their 92 controls and 65 atopic dermatitis patients and their 65 controls. Table 1 shows the characteristics of the sample.

### 3.1. Skin Homeostasis in Psoriatic Patients

Skin barrier function parameters between healthy, involved, and uninvolved skin in psoriatic patients were compared (Figure 1, Appendix A). TEWL was significantly higher at psoriatic plaques (18.45 g·m^−2^·h^−1^) than at uninvolved psoriatic skin (12.06 g·m^−2^·h^−1^) and healthy skin (12.34 g·m^−2^·h^−1^), while no differences were found between uninvolved psoriatic skin and healthy skin. SCH was significantly lower at psoriatic plaques than uninvolved psoriatic skin and healthy skin (8.71 vs. 38.43 vs. 44.39 AU). Temperature was higher at psoriatic plaques than at uninvolved psoriatic skin (30.95 vs. 30.57 °C, *p* = 0.046). The erythema index was significantly higher at psoriatic plaques than at uninvolved psoriatic skin and healthy controls (408.44 vs. 311.56 vs. 285.91 AU). No differences in pH or elasticity were found.

This figure shows TEWL, SCH, temperature, erythema, elasticity, and pH in psoriatic patients, AD patients, and healthy individuals. The differences between healthy skin, uninvolved psoriatic skin, and psoriatic plaque are observed in the left side of each parameter. The differences between healthy skin, uninvolved AD skin, and AD eczematous lesioned skin are found in the right side of each parameter.

The mean PASI was 6.57 (4.82), so patients were divided into two groups: PASI < 7 and PASI ≥ 7 (Table 2). There were no differences in age, sex, or treatment distribution between groups. Regarding current treatment, 27.1% (16/59) patients with PASI < 7 and 21.2% (7/33) patients with PASI ≥ 7 were receiving systemic treatment without differences between groups (*p* = 0.835); 20.3% (12/59) patients with PASI < 7 and 24.2% (8/33) patients with PASI ≥ 7 were receiving biologics, without differences between groups (*p* = 0.941).

SCH was significantly lower in patients with PASI ≥ 7 than in patients with PASI < 7 on psoriatic plaques (4.78 vs. 10.91 AU, *p* < 0.001). Temperature was higher in patients with PASI ≥ 7 than in patients with PASI < 7 on psoriatic plaques (31.56 vs. 30.62 °C, *p* = 0.005). Moreover, it was observed that patients with PASI ≥ 7 had nearly significantly higher TEWL on psoriatic plaques than patients with PASI < 7 (20.75 vs. 17.16 g·m^−2^ h^−1^, *p* = 0.109). There was a negative correlation between SCH on the plaque and PASI (*r* = −0.292, *p* = 0.005), and a nearly significant positive correlation between temperature on the plaque and PASI (*r* = 0.187, *p* = 0.074).

As patients with moderate/severe psoriasis (PASI ≥ 7) exhibited higher temperature values on psoriatic plaques, an ROC curve was generated to determine an optimum cut-off value for temperature that allowed to suspect risk of moderate/severe psoriasis (area under the curve = 0.68, *p* = 0.004). A value for temperature exceeding 30.85 °C indicates, with a sensitivity of 72.7% and a specificity of 55.9%, that a patient had moderate/severe psoriasis. TEWL was also higher in psoriatic patients with moderate/severe PASI; thus, when generating the ROC curve to establish an optimum cut-off point for suspicion of moderate/severe psoriasis (area under the curve = 0.636, *p* = 0.031), it was noted that a TEWL value higher than 13.85 g·m^−2^ h^−1^ indicated that a patient had moderate/severe psoriasis, with a sensitivity of 81.8% and a specificity of 50.8%. SCH was lower in patients with high PASI, so a third ROC curve was generated to establish an optimum cut-off point for this parameter to identify possible patients with a risk of moderate/severe psoriasis (area under the curve = 0.285, *p* = 0.001). A value of SCH lower than 2.07 indicated, with a sensitivity of 60.6% and a specificity of 15.3%, that a patient had moderate/severe psoriasis. Moreover, it was observed that patients with both temperature > 30.85 and TEWL > 13.85 presented moderate/severe psoriasis, with a sensitivity of 60.6% and a specificity of 76.3% (Table 3).

### 3.2. Skin Homeostasis in Atopic Dermatitis Patients

Skin barrier function parameters between healthy, involved, and uninvolved skin in AD patients were compared (Figure 1, Appendix A). TEWL was significantly higher at AD eczematous lesions than at uninvolved AD skin and healthy skin (28.68 vs. 13.15 vs. 11.60 g·m^−2^·h^−1^). SCH was significantly lower at AD eczematous lesions than at uninvolved AD skin and healthy skin (20.20 vs. 40.95 vs. 50.73 AU). Temperature was significantly higher at AD eczematous lesions than at uninvolved AD skin and healthy skin (32.05 vs. 31.35 vs. 31.37 °C), while no differences were found between uninvolved AD skin and healthy skin. The erythema index was significantly higher at AD eczematous lesions than healthy skin (387.21 vs. 244.44 AU). No differences in pH were found. Elasticity was significantly lower at AD eczematous lesions than healthy skin (69% vs. 74% vs. 76%), while no differences were found between uninvolved AD skin and healthy skin.

The mean SCORAD was 36.96 (21.65), so patients were divided into two groups: SCORAD < 37 and SOCRAD ≥ 37 (Table 4). There were no differences in age, sex, or treatment distribution between groups. Regarding current treatment, 30.8% (8/26) patients with SCORAD < 37 and 52.9% (18/34) were receiving systemic treatment without differences between groups (*p* = 0.132). No patient was being treated with biologics.

SCH was significantly lower in patients with SCORAD ≥ 37 than in patients with SCORAD < 37 both at uninvolved AD skin (34.78 vs. 47.10 AU, *p* = 0.003) and AD eczematous lesion (19.90 vs. 30.68 AU, *p* = 0.044). Temperature was higher in patients with SCORAD ≥ 37 than in patients with SCORAD < 37 at AD eczematous lesion (32.45 vs. 31.74, *p* = 0.015). Moreover, it was observed that patients with SOCRAD ≥ 37 had nearly significantly higher TEWL at the AD eczematous lesion than patients with SCORAD < 37 (31.67 vs. 26.33 g·m^−2^·h^−1^, *p* = 0.161). No differences in pH or melanin were found. Elasticity was significantly lower in patients with SCORAD ≥ 37 than in patients with SCORAD < 37, both at uninvolved AD skin (67% vs. 79%, *p* = 0.003) and AD eczematous lesion (63% vs. 75%, *p* = 0.01). Furthermore, a positive correlation between temperature and SCORAD at the AD eczematous lesions (*r* = 0.39, *p* = 0.002) was found, and between TEWL and SCORAD, both at the AD eczematous lesions (*r* = 0.27, *p* = 0.036) and at uninvolved AD skin (*r* = 0.27, *p* = 0.038). A negative correlation between SCH and SCORAD both at the AD eczematous lesions (*r* = −0.364, *p* = 0.005) and at uninvolved AD skin (*r* = −0.519, *p* < 0.001) was observed. Moreover, a negative correlation between elasticity and SCORAD, both at the AD eczematous lesions (*r* = −0.421, *p* = 0.003) and at uninvolved AD skin (*r* = −0.542, *p* < 0.001) was found.

As patients with moderate/severe AD (SCORAD ≥ 37) exhibited higher temperature values at AD eczematous lesions, an ROC curve was generated to determine an optimum cut-off value for temperature that allowed to determine the risk of moderate/severe AD (area under the curve = 0.71, *p* = 0.006). A value for temperature exceeding 31.75 °C indicated, with a sensitivity of 81.8% and a specificity of 57.7%, that a patient had moderate/severe AD. TEWL was also higher in AD patients with moderate/severe SCORAD; thus, when generating the ROC curve to establish an optimum cut-off point for suspicion of moderate/severe AD (area under the curve = 0.633, *p* = 0.078), it was noted that a TEWL value higher than 23.19 g·m^−2^·h^−1^ indicated that a patient had moderate/severe AD, with a sensitivity of 73.5% and a specificity of 53.8%. SCH was lower in patients with high SCORAD, so a third ROC curve was generated to establish an optimum cut-off point for this parameter to identify possible patients with a risk of moderate/severe AD (area under the curve = 0.367, *p* = 0.083). A value of SCH lower than 14.54 AU indicated, with a sensitivity of 71.9% and a specificity of 23.1%, that a patient had moderate/severe AD. Moreover, it was observed that patients with both temperature > 31.75 °C and TEWL > 23.19 g·m^−2^·h^−1^ presented a moderate/severe AD, with a sensitivity of 69.2% and a specificity of 61.8% (Table 5).

### 3.3. Skin Homeostasis Analysis between Psoriatic Patients and AD Patients

It was observed that temperature was higher in AD patients than in psoriatic patients both at uninvolved skin (31.35 vs. 30.56 °C, *p* = 0.001) and involved skin (32.05 vs. 30.95, *p* < 0.001). Moreover, TEWL was higher at eczematous lesions than at psoriatic plaques (28.69 vs. 18.48 g·m^−2^·h^−1^, *p* < 0.001). Erythema was lower at eczematous lesions than at psoriatic plaques (244.50 vs. 311.56, *p* < 0.001). No differences in pH or elasticity were found.

## 4. Discussion

Skin homeostasis analysis showed differences between healthy skin, psoriatic skin, and AD skin. In psoriatic patients, SCH was lower at psoriatic plaques than uninvolved psoriatic skin and healthy controls. Psoriatic plaques showed higher TEWL, temperature, and erythema values than uninvolved psoriatic skin. Temperature and TEWL at psoriatic plaques could help to identify moderate/severe psoriatic patients. In AD patients, TEWL was higher at eczematous lesions than at uninvolved AD skin and healthy controls, while SCH was lower. Eczematous lesions showed higher temperature than uninvolved AD skin. Moreover, AD patients with a more severe disease showed higher temperature, higher TEWL, and lower SCH at their eczematous lesions. Temperature and TEWL at eczematous lesions in AD patients could help to identify AD moderate/severe patients.

This report shows that the whole epidermal barrier is affected in psoriatic patients, not only at psoriatic plaques. Some homeostasis parameters have previously been evaluated in psoriatic patients. Other research showed higher TEWL at psoriatic plaques than at uninvolved psoriatic skin and healthy controls [27,28]. Nevertheless, differences in TEWL values between uninvolved psoriatic skin and healthy controls are controversial [27,28]. Lower SCH values have been found at psoriatic plaques than at uninvolved psoriatic skin and healthy controls, in agreement with our results [15,27]. The differences in TEWL and SCH between psoriatic plaques and uninvolved skin in the same patient could be explained by a decrease in AQP3 expression in plaques and perilesional skin [29]. Controversial results have been reported for pH values. Cannavo et al. found lower pH values for psoriatic skin [15], while Delfino et al. reported no change [30]. Temperature and erythema were also higher at psoriatic skin, explained by its inflammatory pathogenesis [31]. There is a need for reliable assessment of psoriasis severity [32] and, to our knowledge, there is no information regarding a cutaneous homeostasis parameter to assess psoriasis severity. We observed that a value for temperature on psoriatic plaques higher than 30.85 °C indicates, with a sensitivity of 72.7%, that psoriasis is moderate/severe, and that a value for TEWL higher than 13.85 g·m^−2^·h^−1^ indicates, with a sensitivity of 81.8%, that psoriasis is moderate/severe. This may help clinicians to objectively measure psoriasis severity.

Furthermore, this study shows that the whole epidermal barrier is affected in AD patients. TEWL is the most studied parameter in AD patients. Like previous reports, this study shows that TEWL is higher at eczematous AD lesions than at uninvolved AD lesions and healthy skin [33,34,35]. The increased TEWL values reveal an epidermal barrier dysfunction that could be explain by filaggrin mutations [14]. Jungersted et al. also showed that erythema was increases at AD lesions compared to healthy control skin, while SCH was lower and pH was similar at both locations in 49 participants [14]. Moreover, other previous reports, evaluating a smaller number of participants, showed that SCH was higher in healthy controls than at uninvolved AD skin and at eczematous lesions [36]. In agreement with our results, this report shows that the skin barrier function is degraded in AD patients, which is specifically expressed in lesioned skin [36]. This could be explained by a filaggrin deficiency, as this protein is a major constituent of the stratum corneum and contributes to keratin filament aggregation [37]. Temperature and erythema were also higher at eczematous lesions than at uninvolved AD skin and healthy skin, showing inflammatory changes in this disease [38]. To our knowledge, only one previous report has evaluated elasticity parameters in AD patients [39]. Like our results, they observed a more decreased elasticity at AD eczematous lesions than at uninvolved AD skin in 22 patients, without including a healthy control group. Differences in elasticity may reveal that collagen or elastin, the main proteins responsible for skin elasticity [40], are other proteins altered in AD patients.

There is scarce information regarding cutaneous homeostasis parameters and AD severity. Correlations between skin hydration and SCORAD [22,41], and between TEWL and SCORAD [42], have been previously observed. Moreover, it has been shown that TEWL values at non-involved AD skin predicts the development of AD [43,44]. Nevertheless, cut-off points have not been established to assess disease severity. We observed that a value for temperature on the eczematous lesion higher than 31.75 °C indicates, with a sensitivity of 81.8%, that AD is moderate/severe, and that a value for TEWL higher than 23.19 g·m^−2^·h^−1^ indicates, with a sensitivity of 73.5%, that AD is moderate/severe. This research could help clinicians to select AD patients that need to be treated intensively. Moreover, the skin barrier function measurement could also help to resolve the current need for accurate and reproducible scoring systems for the grading of AD [16].

Limitations of this study include the lack of follow-up due to its cross-sectional design, and that patients with different ongoing treatments were included, which might modify epidermal barrier function. Nevertheless, there were no differences in systemic or biologic treatment distribution between severity groups, neither in psoriasis nor in AD.

## 5. Conclusions

In conclusion, the skin barrier is impaired in psoriasis and AD. Temperature and TEWL values may help clinicians to determinate disease severity and select patients who need an intensive treatment.

## Figures and Tables

**Figure 1 jcm-10-00359-f001:**
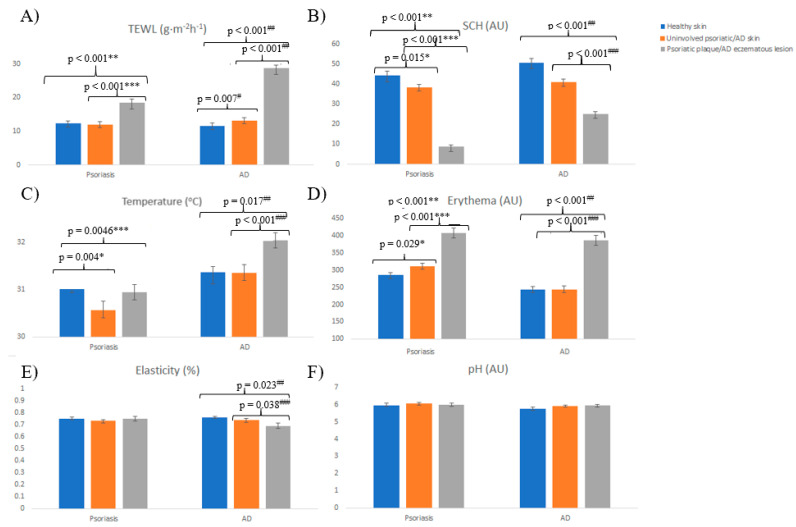
Homeostasis parameters between psoriatic patients and healthy participants and homeostasis parameters between atopic dermatitis patients and healthy participants. (**A**) Transepidermal Water Loss (TEWL) between psoriatic patients and healthy participants and TEWL between atopic dermatitis patients and healthy participants. (**B**) Stratum corneum hydration (SCH) between psoriatic patients and healthy participants and SCH between atopic dermatitis patients and healthy participants. (**C**) Temperature between psoriatic patients and healthy participants and temperature between atopic dermatitis patients and healthy participants. (**D**) Erythema between psoriatic patients and healthy participants and erythema between atopic dermatitis patients and healthy participants. (**E**) Elasticity between psoriatic patients and healthy participants and elasticity between atopic dermatitis patients and healthy participants. (**F**) pH between psoriatic patients and healthy participants and pH between atopic dermatitis patients and healthy participants. Photo caption: AD, atopic dermatitis, AU, arbitrary units, SCH, stratum corneum hydration, TEWL, Transepidermal Water Loss. * *p* value after using Student’s *t* test for independent samples to compare homeostasis parameters between healthy skin and uninvolved psoriatic skin. ** *p* value after using Student’s *t* test for independent samples to compare homeostasis parameters between healthy skin and psoriatic plaque. *** *p* value after using Student’s *t* test for paired samples to compare homeostasis parameters between uninvolved psoriatic skin and psoriatic plaque. ^#^
*p* value after using Student’s *t* test for independent samples to compare homeostasis parameters between healthy skin and uninvolved AD skin. ^##^
*p* value after using Student’s *t* test for independent samples to compare homeostasis parameters between healthy skin and eczematous lesion. ^###^
*p* value after using Student’s *t* test for paired samples to compare homeostasis parameters between AD skin and eczematous lesion.

**Table 1 jcm-10-00359-t001:** Characteristics of the sample. This table shows sociodemographic features in psoriatic patients, atopic dermatitis patients, and healthy participants.

Sociodemographic Features	Psoriatic Patients (*n* = 92)	Healthy Participants assessed on the Elbow (*n* = 92)	Atopic Dermatitis Patients (*n* = 65)	Healthy Participants assessed on the Volar Forearm (*n* = 65)
Age (years)	48.63 (15.70)	42.06 (18.59)	28.14 (19.59)	35.96 (19.03)
Sex (%)				
Female	46 (50%)	57 (62%)	42 (64.6%)	48 (73.8%)
Male	46 (50%)	35 (38%)	23 (35.4%)	17 (26.2%)
Smoking habit (yes)	31 (33.7%)	12 (13%)	7 (10.8%)	6 (9.23%)
Alcohol habit (yes)	29 (31.5%)	29 (31.5%)	15 (23.1%)	10 (15.4%)
Family history of psoriasis/atopic dermatitis (yes)	43 (46.7%)	12 (13%)	32 (49.2%)	7 (10.8%)
Emollients use (yes)	51 (55.4%)	35 (38%)	51 (78.5%)	28 (43.1%)
Treatment				
Topical treatment	49 (53.26%)		39 (60%)	
Systemic treatment	23 (25%)		26 (40%)	
Biologic drugs	20 (21.7%)		0	

Data are expressed as relative (absolute) frequencies and means (standard deviations (SDs)).

**Table 2 jcm-10-00359-t002:** Homeostasis parameters in psoriatic patients depending on disease severity. This table shows differences in TEWL, SCH, temperature, erythema, melanin, pH, and elasticity between patients with mild psoriasis (PASI < 7) and patients with moderate/severe psoriasis (PASI ≥ 7).

Skin Homeostasis Parameters	Psoriatic Patients with PASI < 7 (*n* = 59)		Psoriatic Patients with PASI ≥ 7 (*n* = 33)		*p* Value	*p* Value
	Uninvolved Psoriatic Skin	Psoriatic Plaques	Uninvolved Psoriatic Skin	Psoriatic Plaques	*p* *	*p* **
TEWL (g·m^−2^·h^−1^)	12.18 (7.52)	17.16 (9.58)	11.86 (8.78)	20.75 (11.22)	0.855	0.109
SCH (AU)	37.76 (13.13)	10.91 (9.76)	39.63 (14.69)	4.78 (5.24)	0.531	<0.001 **
Temperature (°C)	30.51 (2.00)	30.62 (1.65)	30.66 (1.09)	31.56 (1.13)	0.639	0.005 **
Erythema (AU)	311.78 (73.15)	404.37 (73.76)	311.34 (69.90)	412.79 (67.91)	0.981	0.648
Melanin (AU)	246.49 (81.63)	193.65 (69.98)	230.05 (75.64)	188.36 (69.30)	0.422	0.770
pH	6.01 (0.64)	6.06 (1.01)	6.12 (0.56)	5.90 (0.87)	0.422	0.468
Elasticity (%)	0.74 (0.13)	0.77 (0.20)	0.69 (0.15)	0.72 (0.17)	0.110	0.251

AU, arbitrary units; PASI, psoriasis area and severity index; SCH, stratum corneum hydration; TEWL, transepidermal water loss; * *p*-value after using Student’s *t*-test for independent samples to compare homeostasis parameters between uninvolved psoriatic skin in psoriatic patients with PASI < 7 and uninvolved psoriatic skin in psoriatic patients with PASI ≥ 7; ** *p*-value after using Student’s *t*-test for independent samples to compare homeostasis parameters between psoriatic plaques in psoriatic patients with PASI < 7 and psoriatic plaques in psoriatic patients with PASI ≥ 7.

**Table 3 jcm-10-00359-t003:** Odds ratios for main parameters analyzed in the study to predict moderate/severe psoriasis (PASI ≥ 7). Sensitivity and specificity values to predic moderate/severe psoriasis based on skin homeostasis parameters, cut-off values, and odds ratios.

Skin Homeostasis Parameters	Cut-off Value	Sensitivity	Specificity	OR	*p*
Temperature (°C)	30.85	72.7%	55.9%	3.39	0.010 *
TEWL (g·m^−2^ h^−1^)	13.85	81.8%	50.8%	4.66	0.003 *
SCH (AU)	2.07	39.4%	84.7%	0.28	0.011 *
Two criteria (temperature > 30.85 + TEWL > 13.85)	-	60.6%	76.3%	4.95	0.001 *

AU, arbitrary units; OR, odds ratio; PASI, psoriasis area and severity index; SCH, stratum corneum hydration; TEWL, transepidermal water loss. * *p* value after using a logistic regression to evaluate the association between disease severity (independent variable), as a categoric variable (PASI < 7 or PASI ≥ 7) and each skin homeostasis parameter cut-off point (dependent variable), considered as a categoric variable (lower o equal than the cut-off point o higher than the cut-off point).

**Table 4 jcm-10-00359-t004:** Homeostasis parameters in atopic dermatitis patients depending on disease severity. This table shows differences in TEWL, SCH, temperature, erythema, melanin, pH, and elasticity between patients with mild AD (SCORAD < 37) and patients with moderate/severe (SCORADI ≥ 37).

Skin Homeostasis Parameters	AD Patients with SCORAD < 37 (*n* = 26)		AD Patients with SCORAD ≥ 37 (*n* = 34)		*p* Value	*p* Value
	Uninvolved AD Skin	AD Eczematous Lesion	Uninvolved AD Skin	AD Eczematous Lesion	*p* *	*p* **
TEWL (g·m^−2^ h^−1^)	10.88 (8.04)	26.33 (15.34)	13.75 (6.62)	31.67 (13.74)	0.135	0.161
SCH (AU)	47.10 (17.01)	30.68 (24.23)	34.78 (13.55)	19.90 (11.40)	0.003 *	0.044 **
Temperature (°C)	31.30 (1.04)	31.74 (1.00)	31.35 (1.46)	32.45 (1.15)	0.891	0.015 **
Erythema (AU)	201.05 (17.30)	351.78 (102.64)	254.93 (78.78)	395.71 (77.71)	0.004 *	0.361
Melanin (AU)	168.91 (37.39)	1990.04 (23.33)	212.55 (82.57)	215.24 (88.88)	0.221	0.298
pH	5.79 (0.62)	5.87 (0.61)	6.04 (0.41)	6.03 (0.47)	0.97	0.274
Elasticity (%)	0.79 (0.11)	0.75 (0.12)	0.67 (0.16)	0.63 (0.20)	0.003 *	0.01 **

AD, atopic dermatitis; AU, arbitrary units; PASI, psoriasis area and severity index; SCH, stratum corneum hydration; TEWL, transepidermal water loss; * *p*-value after using Student’s *t*-test for independent samples to compare homeostasis parameters between uninvolved AD skin in AD patients with SCORAD < 37 and uninvolved AD skin in AD patients with SCORAD ≥ 37; ** *p*-value after using Student’s *t*-test for independent samples to compare homeostasis parameters between AD eczematous lesion in AD patients with SCORAD < 37 and AD eczematous lesion in AD patients with SCORAD ≥ 37.

**Table 5 jcm-10-00359-t005:** Odds ratios for main parameters analyzed in the study to predict moderate/severe AD (SCORAD > 37). Sensitivity and specificity values to predic moderate/severe AD based on skin homeostasis parameters, cut-off values, and odds ratio.

Skin Homeostasis Parameters	Cut-off Value	Sensitivity	Specificity	OR	*p*
Temperature (°C)	31.75	81.8%	57.7%	6.14	0.003 *
TEWL (g·m^−2^ h^−1^)	23.19	73.5%	53.8%	3.24	0.034 *
SCH (AU)	14.54	71.9%	23.1%	0.77	0.663
Two criteria (temperature > 31.75 + TEWL > 23.19)	-	69.2%	61.8%	3.64	0.19

AU, arbitrary units; OR, odds ratio; SCH, stratum corneum hydration; SCORAD, SCORing Atopic Dermatitis; TEWL, transepidermal water loss. * *p* value after using a logistic regression to evaluate the association between disease severity (independent variable), as a categoric variable (SCORAD < 37 or SCORAD ≥ 37) and each skin homeostasis parameter cut-off point (dependent variable), considered as a categoric variable (lower o equal than the cut-off point o higher than the cut-off point).

## Data Availability

The data presented in this study are available on request from the corresponding author.

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
