# Peer review of "Skin Barrier Function in Psoriasis and Atopic Dermatitis: Transepidermal Water Loss and Temperature as Useful Tools to Assess Disease Severity"

_jcm, 2021, doi:10.3390/jcm10020359_

Round 1
Reviewer 1 Report
The authors (Montero-Vilchez , et al.) focused on Trans-epidermal Water Loss and Temperature as useful tools to assess disease severity such as psoriasis and dermatitis. Whole manuscript is well summarized, but still have some points to be addressed or cleared.
I. Some sentences through out the manuscript are grammatically incorrect, difficult to interpret and often incomplete or repetitive as important details are missing. Some examples but not all are given below. Also, punctuation requires attention.
line 17. dermatitis (AD) severity but most of them are based of subjective. Instead of "based of" it is better to change it to "based on".
line 19. "to accurately evaluate disease severity" can be changed to "to evaluate disease severity accurately".
line 29. temperature>31.75ºC presented a moderate/severe AD (SCORAD≥37) with a with a sensitivity of 81.8%. There is redundancy: "with a" is repeated 2 times.
line 30. conclusion, temperature and TEWL values may help clinicians to determinate disease severity and. it is better to say determine not determinate.
line 32. select patients who need an intensive treatment. Remove the article "a".
line 44-46. Psoriasis and atopic dermatitis (AD) are both cutaneous inflammatory diseases resulting of the interaction between both environmental and genetic factors that may alter epidermal barrier function[11]. This sentence needs to be rephrased as following; Psoriasis and atopic dermatitis (AD) are cutaneous inflammatory diseases resulting from the interaction between environmental and genetic factors that may alter epidermal barrier function [11].
line 66. psoriatic and AD patients could help clinicians to assess objectively the disease severity[24]. it is better to say to assess the disease severity objectively.
line 302-304. Skin homeostasis analysis showed differences between healthy skin, psoriatic skin and AD skin. In psoriatic patients, SCH was lower at psoriatic plaques than uninvolved psoriatic skin and healthy controls and psoriatic plaques showed higher TEWL, temperature.. Please check the punctuation here. Commas are missing. This is an example but try to check the punctuation overall the manuscript.
II. The figures and the tables need a short description just after the title.
Author Response
Some sentences through out the manuscript are grammatically incorrect, difficult to interpret and often incomplete or repetitive as important details are missing. Some examples but not all are given below. Also, punctuation requires attention.
Thank your very much for your comments. The manuscript has been thoroughly reviewed to avoid these mistakes.
line 17. dermatitis (AD) severity but most of them are based of subjective. Instead of "based of" it is better to change it to "based on".
“Based of” has been changed to “based on”.
line 19. "to accurately evaluate disease severity" can be changed to "to evaluate disease severity accurately".
“To accurately evaluate disease severity” has been changed to “to evaluate disease severity accurately”.
line 29. temperature>31.75ºC presented a moderate/severe AD (SCORAD≥37) with a with a sensitivity of 81.8%. There is redundancy: "with a" is repeated 2 times.
“With a” has been removed once.
line 30. conclusion, temperature and TEWL values may help clinicians to determinate disease severity and. it is better to say determine not determinate.
The word “determinate” has been changed to “determine”.
line 32. select patients who need an intensive treatment. Remove the article "a".
The article has been removed.
line 44-46. Psoriasis and atopic dermatitis (AD) are both cutaneous inflammatory diseases resulting of the interaction between both environmental and genetic factors that may alter epidermal barrier function[11]. This sentence needs to be rephrased as following; Psoriasis and atopic dermatitis (AD) are cutaneous inflammatory diseases resulting from the interaction between environmental and genetic factors that may alter epidermal barrier function [11].
This sentence has been rephrased as recommended.
line 66. psoriatic and AD patients could help clinicians to assess objectively the disease severity[24]. it is better to say to assess the disease severity objectively.
“To assess objectively the disease severity” has been changed to “to assess the disease severity objectively”.
line 302-304. Skin homeostasis analysis showed differences between healthy skin, psoriatic skin and AD skin. In psoriatic patients, SCH was lower at psoriatic plaques than uninvolved psoriatic skin and healthy controls and psoriatic plaques showed higher TEWL, temperature.. Please check the punctuation here. Commas are missing. This is an example but try to check the punctuation overall the manuscript.
Punctuation has been corrected in this sentence and has been reviewed throughout the manuscript.
The figures and the tables need a short description just after the title.
In the figures and tables, a description after the title has been added.

Reviewer 2 Report
This is an interesting well written manuscript evaluating the alterations of skin barrier function in AD skin and psoriatic skin, compared to the skin of healthy controls.
Authors well described the methods used to evaluate these alterations, and their conclusions are well supported by their results.
There are only few issues that should be clarified:
- psoriatic and atopic dermatitis patients suffering from most severe forms ( PASI> 7 and SOCRAD≥37 respectively) were under systemic treatments?
- Were any patients treated with biologics? Please specify ongoing systemic treatments for both groups.
- Different ongoing treatments may modify psoriasis activity as well as some parameters (such as skin temperature) influencing the final rilevations. This aspect should be reported in the study limitations.
Author Response
Reviewer 2
This is an interesting well written manuscript evaluating the alterations of skin barrier function in AD skin and psoriatic skin, compared to the skin of healthy controls.
Authors well described the methods used to evaluate these alterations, and their conclusions are well supported by their results.
Thank you very much for your comments.
There are only few issues that should be clarified:
psoriatic and atopic dermatitis patients suffering from most severe forms ( PASI> 7 and SOCRAD≥37 respectively) were under systemic treatments?
Some of them were receiving systemic treatments but others weren’t. Regarding psoriatic patients, 27.1% (16/59) patients with PASI<7 and 21.2% (7/33) patients with PASI≥7 were receiving systemic treatment, without differences between groups (p=0.835). Regarding atopic dermatitis patients, 30.8% (8/26) patients with SCORAD<37 and 52.9% (18/34) were receiving systemic treatment without differences between groups (p=0.132). This information has been added to its corresponding paragraph and the overall treatment rate has been added to table 1.
Were any patients treated with biologics? Please specify ongoing systemic treatments for both groups.
None of atopic dermatitis patients were treated with biologics. Regarding psoriatic patients, 20.3% (12/59) patients with PASI<7 and 24.2% (8/33) patients with PASI≥7 were receiving biologics, without differences between groups (p=0.941). This information has been added to the table 1. This information has been added to its corresponding paragraph and the overall treatment rate has been added to table 1.
Different ongoing treatments may modify psoriasis activity as well as some parameters (such as skin temperature) influencing the final rilevations. This aspect should be reported in the study limitations.
This aspect has been added to the study limitations. It has also been added that there were no differences regarding treatment between groups.
